# COVID-19 pandemic and trends in new diagnosis of atrial fibrillation: A nationwide analysis of claims data

**Inmaculada Hernandez**[1]*, **Meiqi He**[1], **Jingchuan Guo**[2], **Mina Tadrous**[3], **Nico Gabriel**[1], **Gretchen Swabe**[4], **Walid F. Gellad**[5], **Utibe R. Essien**[5], **Samir Saba**[4], **Emelia J. Benjamin**[6], **Jared W. Magnani**[4]

1 Division of Clinical Pharmacy, University of California, San Diego, Skaggs School of Pharmacy and Pharmaceutical Sciences, La Jolla, California, United States of America, 2 Department of Pharmaceutical Outcomes and Policy, University of Florida College of Pharmacy, Gainesville, Florida, United States of America, 3 Leslie Dan Faculty of Pharmacy, University of Toronto, Toronto, Canada, 4 Division of Cardiology, Department of Medicine, University of Pittsburgh School of Medicine, Pittsburgh, Pennsylvania, United States of America, 5 Division of General Internal Medicine, University of Pittsburgh School of Medicine, Pittsburgh, PA and Center for Health Equity Research and Promotion, VA Pittsburgh Healthcare System Pittsburgh, Pennsylvania, United States of America, 6 Department of Medicine, Boston Medical Center and Boston University Department of Medicine, Chobanian and Avedisian School of Medicine, and Department of Epidemiology, School of Public Health, Boston, Massachusetts, United States of America

* inhernandez@health.ucsd.edu

## Abstract

### Background

Atrial fibrillation (AF) is associated with a five-fold increased risk of stroke and a two-fold increased risk of death. We aimed to quantify changes in new diagnoses of AF following the onset of the COVID-19 pandemic. Investigating changes in new diagnoses of AF is of relevance because delayed diagnosis interferes with timely treatment to prevent stroke, heart failure, and death.

### Methods

Using De-identified Optum's Clinformatics® Data Mart, we identified 19,500,401 beneficiaries continuously enrolled for 12 months in 2016-Q3 2020 with no history of AF. The primary outcome was new AF diagnoses per 30-day interval. Secondary outcomes included AF diagnosis in the inpatient setting, AF diagnosis in the outpatient setting, and ischemic stroke as initial manifestation of AF. We constructed seasonal autoregressive integrated moving average models to quantify changes in new AF diagnoses after the onset of the COVID-19 pandemic (3/11/2020, date of pandemic declaration). We tested whether changes in the new AF diagnoses differed by race and ethnicity.

### Results

The average age of study participants was 51.0±18.5 years, and 52% of the sample was female. During the study period, 2.7% of the study sample had newly-diagnosed AF. New AF diagnoses decreased by 35% (95% CI, 21%-48%) after the onset of the COVID-19

**Data Availability Statement:** Optum's Clinformatics® Data Mart (CDM) is derived from a database of administrative health claims for members of large commercial and Medicare

Advantage health plans. Clinformatics® Data Mart is statistically de-identified under the Expert Determination method consistent with HIPAA and managed according to Optum® customer data use agreements. CDM administrative claims submitted for payment by providers and pharmacies are verified, adjudicated and de-identified prior to inclusion. This data, including patient-level enrollment information, is derived from claims submitted for all medical and pharmacy health care services with information related to health care costs and resource utilization. The population is geographically diverse, spanning all 50 states. Optum's Clinformatics® Data Mart are licensed and were accessed for the study under a data user agreement that does not allow data sharing. The authors obtained access to the data under a data user agreement. Other investigators engaging in a similar license and data user agreement would be able to access the same data if they purchased access for the same years of data and files. Investigators not engaging in a license and data user agreement would not be able to access the data as the data are not publicly available. Investigators interested in obtaining access can contact optum at 1-866-306-1321 or connected@optum.com.

**Funding:** This work was funded by the National Heart, Lung and Blood Institute (grants K01HL142847 and R01HL15705). Dr. Benjamin is funded by the National Heart, Lung and Blood Institute (R01HL092577); and the American Heart Association (AHA_18SFRN34110082). Dr. Magnani is funded by the National Heart, Lung and Blood Institute (R33HL144669 and R01HL143010). The funders had no role in study design, data collection and analysis, decision to publish, or preparation of the manuscript.

**Competing interests:** I have read the journal's policy and the authors of this manuscript have the following competing interests: Hernandez has received consulting fees from Pfizer and Bristol Myers Squibb, outside of the submitted work. Our conflicts do not alter our adherence to PLOS ONE policies on sharing data and materials.

pandemic, from 1.14 per 1000 individuals (95% CI, 1.05–1.24) to 0.74 per 1000 (95% CI, 0.64 to 0.83, p-value<0.001). New AF diagnoses decreased by 37% (95% CI, 13%- 55%) in the outpatient setting and by 29% (95% CI, 14%-43%) in the inpatient setting. The decrease in new AF diagnoses was similar across racial and ethnic subgroups.

## Conclusion

In a nationwide cohort of 19.5 million individuals, new diagnoses of AF decreased substantially following the onset of the COVID-19 pandemic. Our findings evidence pandemic disruptions in access to care for AF, which are concerning because delayed diagnosis interferes with timely treatment to prevent complications.

## Introduction

Disruptions in access to care associated with the COVID-19 pandemic represent potentially important health impacts of the pandemic beyond COVID cases and deaths [1]. An emerging body of literature has reported decreases in the number of patients seeking emergency care in the earlier months of the COVID-19 pandemic [2–5]. Significant reductions also have been observed in outpatient care for patients with chronic disease, even after accounting for the increased uptake of telemedicine [6]. Other reports have documented decreased provision of imaging, laboratory services, screening, surgical interventions, or provider-administered drugs for non-COVID disease [7–9]. These prior studies constitute important contributions to the understanding of disruptions in health care for non-COVID disease during the COVID-19 pandemic. However, because they focused on measuring rates of events as opposed to following a population cohort over time, prior investigations do not allow for a differentiation of disruptions in care for new-onset versus existing disease.

Evaluating pandemic distortions of diagnosis of new-onset chronic disease is important because delayed diagnosis may interfere with treatment, resulting in worsened prognosis. This is particularly relevant for diseases with life-threatening complications that can be prevented through medical or pharmacological interventions. One such example is atrial fibrillation (AF), the most common cardiac arrhythmia globally [10, 11]. AF is associated with a five-fold increased risk of stroke and two-fold increased risk of death [12]. Oral anticoagulation is available for patients diagnosed with AF and reduces the risk of stroke by over 60% and death by over 20% [13]. AF is a critical disease state to measure the effects of the COVID-19 pandemic on non-COVID related chronic disease because every aspect of stroke prevention is vulnerable to disruption, including diagnosis, initiation of anticoagulation therapy, and treatment monitoring.

We aimed to quantify changes in new diagnoses of AF following the onset of the COVID-19 pandemic. Because the COVID-19 pandemic disproportionately affected underrepresented racial/ethnic groups, it was relevant to evaluate whether changes in diagnoses following pandemic declaration differed across racial/ethnic groups.

## Methods

### Data source and study population

We obtained 1/1/2016-9/30/2020 de-identified data from Optum's Clinformatics® Data Mart, which is derived from a database of administrative health claims for members of large

commercial and Medicare Advantage health plans. These data include verified, adjudicated and de-identified medical and pharmacy claims for a geographically diverse population spanning all 50 states in the U.S. We selected the study population in four steps (**S1 Fig**): First, we selected patients who were continuously enrolled for at least 12 months in 1/1/2016-3/30/2020. Second, we constrained sampling to those older than 18 years of age. Third, we excluded patients who had a diagnosis of AF in the first 12 months of continuous enrollment (definition of AF is listed under the outcomes section). We used this 12-month washout to exclude individuals who represent patients with prevalent AF. Finally, we excluded patients with incomplete covariate information (2.4% of the sample). The final sample included 19,500,401 individuals free of AF on index date, which was defined as the first day after completing the 12-month washout period. Patients were followed until the first of the following events: death, disenrollment, diagnosis of AF, or end of the study (9/30/2020). The Institutional Review Board at the University of California, San Diego approved this study as exempt.

## Outcomes

The primary outcome was new AF diagnosis, which was defined as having an inpatient or outpatient claim with International Classification of Diseases Ninth Revision (ICD-9) code 427.31 or International Classification of Diseases Tenth Revision (ICD-10) codes I48.0, I48.1, I48.2, or I48.91 in the first or second diagnosis field [14]. Previous studies have estimated the positive predictive value of our definition to be 93–97% [15, 16]. We defined the primary outcome as one inpatient or one outpatient diagnosis as opposed to one inpatient or two outpatient diagnoses [14, 17–19] for two reasons: first, the objective was to measure changes in new AF diagnoses after the COVID-19 pandemic, and the requirement for two outpatient diagnosis would have affected the ability to detect immediate changes after the pandemic declaration due to the time lapse between first and second diagnosis, second, the requirement for second outpatient diagnosis would have affected the identification of ischemic stroke as initial manifestation of AF.

Secondary outcomes included AF diagnosis in the inpatient setting, AF diagnosis in the outpatient setting, and ischemic stroke as initial manifestation of AF. The setting of AF diagnosis was ascertained using the place of service in the claim with the first recorded AF diagnosis; places of service with code 21 (inpatient hospital) were categorized as inpatient. Diagnoses that did not originate from the inpatient setting were categorized as outpatient. Ischemic stroke as initial manifestation of AF was defined as having a stroke event on the day or in the 30 days prior to the first AF diagnosis, as previously reported in the literature [20]. The stroke event was identified using inpatient claims and ICD-10 diagnosis code I63.

## Independent variables

The main independent variable of interest was time after the World Health Organization declaration of pandemic (3/11/2020). Covariates included age, sex, race, ethnicity, and state and were identified as of index date. Race and ethnicity were categorized into non-Hispanic White, non-Hispanic Black, Hispanic, and other, which included Asian and unknown. In Optum, race and ethnicity data are collected using public records and imputation with commercial software that uses algorithms developed with census data and first and last names. This method has 71% positive predictive value for estimating Black race [21].

## Statistical analysis

We determined the rate of AF diagnosis for each 30-day interval during the study period as the quotient between the number of patients who had a first AF diagnosis in the given time interval (numerator) and the population who remained at risk in each 30-day interval

(denominator). An individual was included in the denominator after the 12-month washout period and was censored and excluded from the denominator at death, disenrollment, or first AF diagnosis.

We constructed seasonal autoregressive integrated moving average (ARIMA) models that accounted for seasonality to test changes in the level of diagnosis after pandemic declaration (intercept) and the trend of diagnosis after pandemic declaration (slope). We added an indicator variable for the period after 03/11/2020 and a ramp function to specify the increase in intervals after 03/11/2020. ARIMA models were selected because this technique accommodates the seasonality and autocorrelation commonly found in time-series of diagnosis and healthcare data [22, 23]. ARIMA models included autoregressive terms, moving average terms, the differences of raw values, and seasonal terms with a period set to 12 months. We reported observed results and values predicted with ARIMA models for the overall sample and for subgroups defined by gender and race, and ethnicity. We also used ARIMA models to predict trends in AF diagnosis in the absence of the COVID-19 pandemic.

We conducted interrupted time series analyses with linear regression to formally test whether changes in outcomes after pandemic declaration differed by subgroup [24], represented with a three-way interaction between subgroup, continuous time, and the indicator for the post-pandemic period. Interrupted time series analyses with linear regression models were necessary to formally test whether changes in outcomes after pandemic declaration differed by subgroup because ARIMA models are not able to accommodate this interaction analysis.

Finally, we investigated state variation in the changes in new AF diagnoses by comparing the rates of diagnosis in the 30-day interval before versus after pandemic declaration. We reported results for states with at least 200,000 study participants to ensure stability of estimates. We examined whether changes in new AF diagnosis were related to the incidence of COVID-19 cases per 100,000 state residents as of 4/9/2020. COVID case data were obtained from the Centers for Disease Control and Prevention COVID Data Tracker [25]. Two-tailed p value <0.05 was defined as significant. Analyses were conducted using SAS 9.4 (Cary, NC).

## Results

### Study sample

The cohort included 19,500,401 individuals free of AF on the index date. The cohort was 52% women with nearly half <50 years of age, 18.3% age 65–74, and 11.1% age≥75 years (**Table 1**). Non-Hispanic White individuals comprised the majority of the cohort (59.8%) followed by Hispanic individuals (11.7%), and non-Hispanic Black individuals (9.2%). Medicare beneficiaries accounted for 30.9% of study participants.

Across the study period, 519,950 individuals or 2.7% of the cohort were newly diagnosed with AF. As expected, study participants who developed AF during follow-up had a higher average age than those who did not and were more likely to be Medicare beneficiaries.

### Changes in new atrial fibrillation diagnoses

The 30-day incidence rate of new AF diagnoses presented a seasonal pattern, with higher incidence in winter months (**Fig 1**). Following the COVID-19 pandemic declaration, AF diagnoses were estimated to decrease by 35% (95% CI, 21%-48%), from 1.14 per 1000 individuals (95% CI, 1.05 to 1.24) before the onset of the COVID-19 pandemic to 0.74 per 1000 (95% CI, 0.64 to 0.83) after pandemic onset (p-value for level of change<0.001), **Fig 1** and **S1 Table.** The observed incidence of new AF diagnosis was substantially lower than predicted by regression models in the first 90 days after the declaration of the COVID-19 pandemic, however these

**Table 1. Baseline patient characteristics.**

| Variable | Total Population (n = 19,500,401) | Not Diagnosed with AF (n = 18,980,451, 97.3%) | Diagnosed with AF (n = 519,950, 2.7%) |
|---|---|---|---|
| Female, No. (%) | 10,147,297 (52.0) | 9,898,009 (52.1) | 249,288 (47.9) |
| Age, Mean years (Std.) | 51.02 (18.5) | 50.40 (18.3) | 73.50 (11.0) |
| Age, years | | | |
| <50, No. (%) | 9,284,245 (47.6) | 9,266,428 (48.8) | 17,817 (3.4) |
| 50–64, No. (%) | 4,481,400 (23.0) | 4,413,175 (23.3) | 68,225 (13.1) |
| 65–74, No. (%) | 3,571,769 (18.3) | 3,401,055 (17.9) | 170,714 (32.8) |
| ≥75, No. (%) | 2,162,987 (11.1) | 1,899,793 (10.0) | 263,194 (50.6) |
| Race and Ethnicity | | | |
| Non-Hispanic White, No. (%) | 11,657,821 (59.8) | 11,305,424 (59.6) | 352,397 (67.8) |
| Non-Hispanic Black, No. (%) | 1,788,365 (9.2) | 1,740,517 (9.2) | 47,848 (9.2) |
| Hispanic, No. (%) | 2,282,787 (11.7) | 2,239,340 (11.8) | 43,447 (8.4) |
| Other, No. (%) | 3,771,428 (19.3) | 3,695,170 (19.5) | 76,258 (14.7) |
| Medicare, No. (%) | 6,035,047 (30.9) | 5,596,390 (29.5) | 438,657 (84.4) |

Abbreviations: AF, atrial fibrillation.

differences narrowed by summer 2020. As of June 2020, the observed incidence of new AF diagnoses was only 6% lower than predicted.

On average, in the pre-pandemic period, ischemic stroke as initial manifestation of AF accounted for 4.4% of all new AF diagnoses. Following the COVID-19 pandemic declaration, the rate of ischemic stroke as initial manifestation of AF was estimated to decrease by 31% (95% CI, 4%-51%), from 0.055 per 1000 (95% CI, 0.048 to 0.063) before the onset of the COVID-19 pandemic to 0.038 per 1000 after pandemic onset (95% CI, 0.031 to 0.046), p-value for level of change<0.001.

## Changes in new atrial fibrillation diagnoses by setting

**Fig 2** shows the trend in the 30-day incidence rate of new AF diagnoses by setting of diagnosis. Following the onset of the COVID-19 pandemic, AF diagnoses were estimated to decrease by 37% (95% CI,13%- 55%) in the outpatient setting, from 0.74 per 1000 before pandemic declaration to 0.47 per 1000 after (p-value for level of change<0.001) and by 29% (95% CI, 14%-43%) in the inpatient setting, from 0.39 per 1000 before pandemic declaration to 0.27 per 1000 after (p-value for level of change<0.001), **Fig 2** and **S1 Table.**

The upper panel shows trends in new atrial fibrillation diagnoses per 1000 individuals in the inpatient setting for 30-day intervals, from 01/02/2016 to 09/06/2020. The lower panel shows trends in new atrial fibrillation diagnoses per 1000 individuals in the outpatient setting for 30-day intervals, from 01/02/2016 to 09/06/2020. Solid red lines represent observed values. Solid blue lines represent predictions from ARIMA models the absence of the COVID-19 pandemic. In other words, they represent trends in AF diagnosis that would have been observed if there had not been a change in AF diagnoses following the COVID-19 pandemic. Dashed blue lines represent confidence intervals.

## Changes in new atrial fibrillation diagnoses by subgroup

In the 30-day period preceding the onset of the COVID-19 pandemic (2/10/2020-3/10/2020), the estimated 30-day incidence rate of new AF diagnoses per 1000 individuals averaged 1.22 for White individuals, 1.15 for Black individuals, 0.82 for Hispanic individuals, and 1.05 for

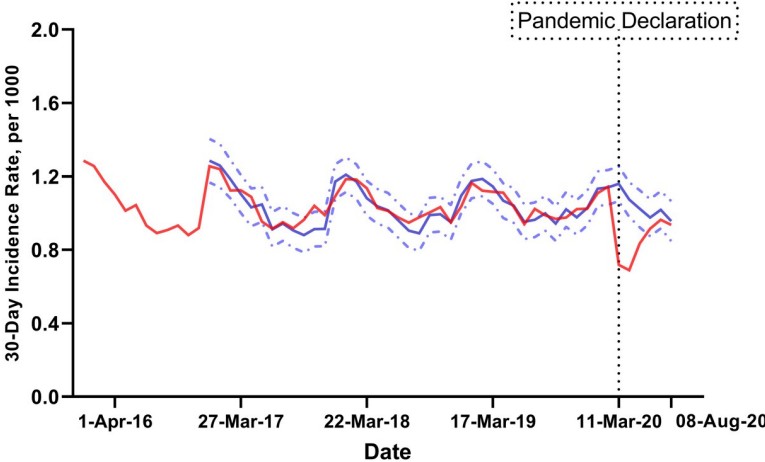

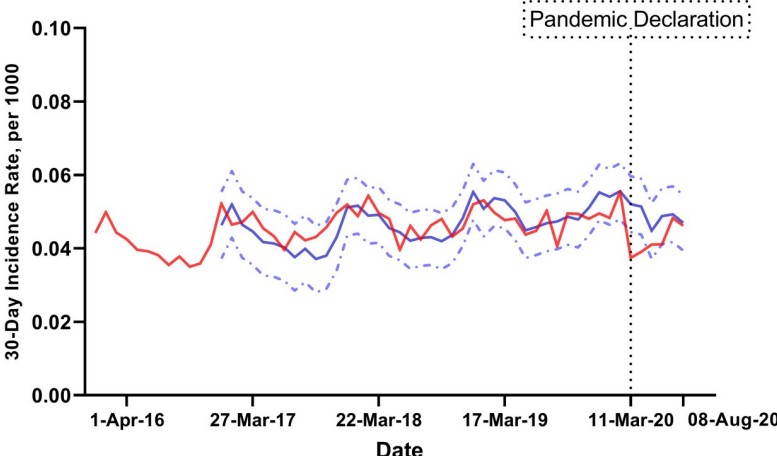

**Fig 1. Observed and predicted new atrial fibrillation diagnoses, overall sample.** Abbreviations: UCL, upper confidence limit; LCL, lower confidence limit. The upper panel shows trends in new atrial fibrillation diagnoses per 1000 individuals for 30-day intervals, from 01/02/2016 to 09/06/2020. The lower panel shows trends in ischemic strokes as initial manifestation of atrial fibrillation per 1000 individuals for 30-day intervals, from 01/02/2016 to 09/06/2020. Solid red lines represent observed values. Solid blue lines represent predictions from ARIMA models the absence of the COVID-19 pandemic. In other words, they represent trends in AF diagnosis that would have been observed if there had not been a change in AF diagnoses following the COVID-19 pandemic. Dashed blue lines represent confidence intervals.

individuals of other races or ethnicities (**Fig 3**). Following the onset of the COVID-19 pandemic, the estimated decrease in new AF diagnoses was significant across all racial and ethnic subgroups, but the magnitude of the decrease did not differ across racial or ethnic subgroups (p = 0.34), as shown in **Fig 3 and S2** and **S3 Tables**.

The upper left panel shows trends in new atrial fibrillation diagnoses per 1000 individuals in White individuals for 30-day intervals, from 01/02/2016 to 09/06/2020. The upper right panel shows trends in new atrial fibrillation diagnoses per 1000 individuals in Black

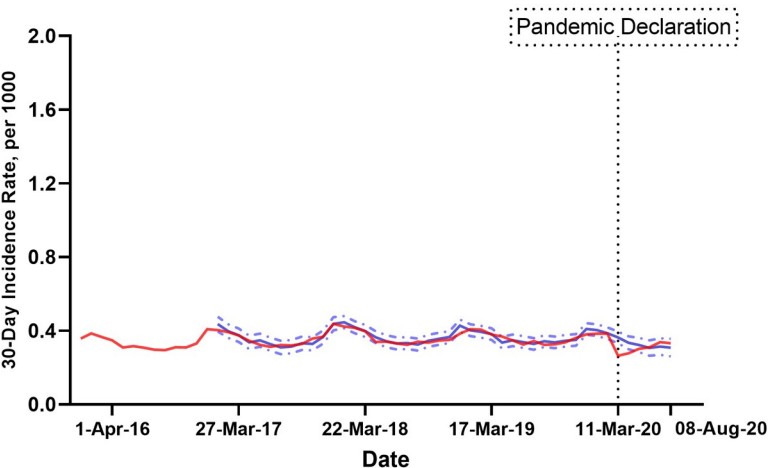

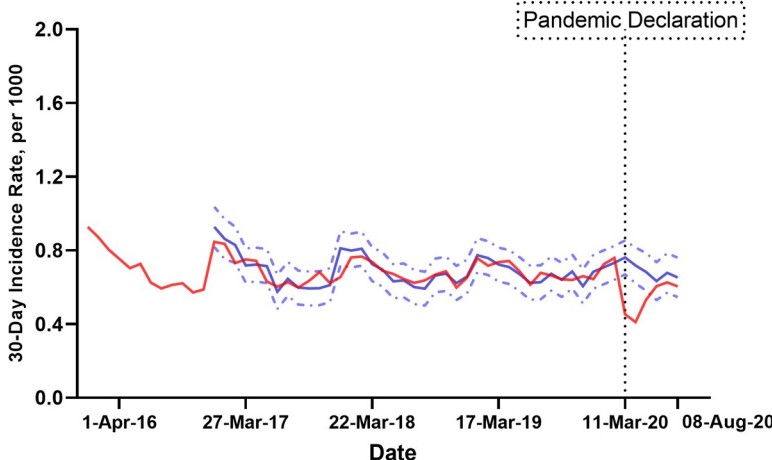

**Fig 2. Observed and predicted new atrial fibrillation diagnoses by setting.** Abbreviations: UCL, upper confidence limit; LCL, lower confidence limit.

individuals for 30-day intervals, from 01/02/2016 to 09/06/2020. The lower left panel shows trends in new atrial fibrillation diagnoses per 1000 individuals in Hispanic individuals for 30-day intervals, from 01/02/2016 to 09/06/2020. The lower right panel shows trends in new atrial fibrillation diagnoses per 1000 individuals in individuals of other races for 30-day intervals, from 01/02/2016 to 09/06/2020. Solid red lines represent observed values. Solid blue lines represent predictions from ARIMA models the absence of the COVID-19 pandemic. In other words, they represent trends in AF diagnosis that would have been observed if there had not been a change in AF diagnoses following the COVID-19 pandemic. Dashed blue lines represent confidence intervals.

The estimated decrease in new AF diagnoses was significant across both sex subgroups, but the magnitude of the decrease did not differ across sex groups, as shown in **S2 Fig** and **S4 Table**.

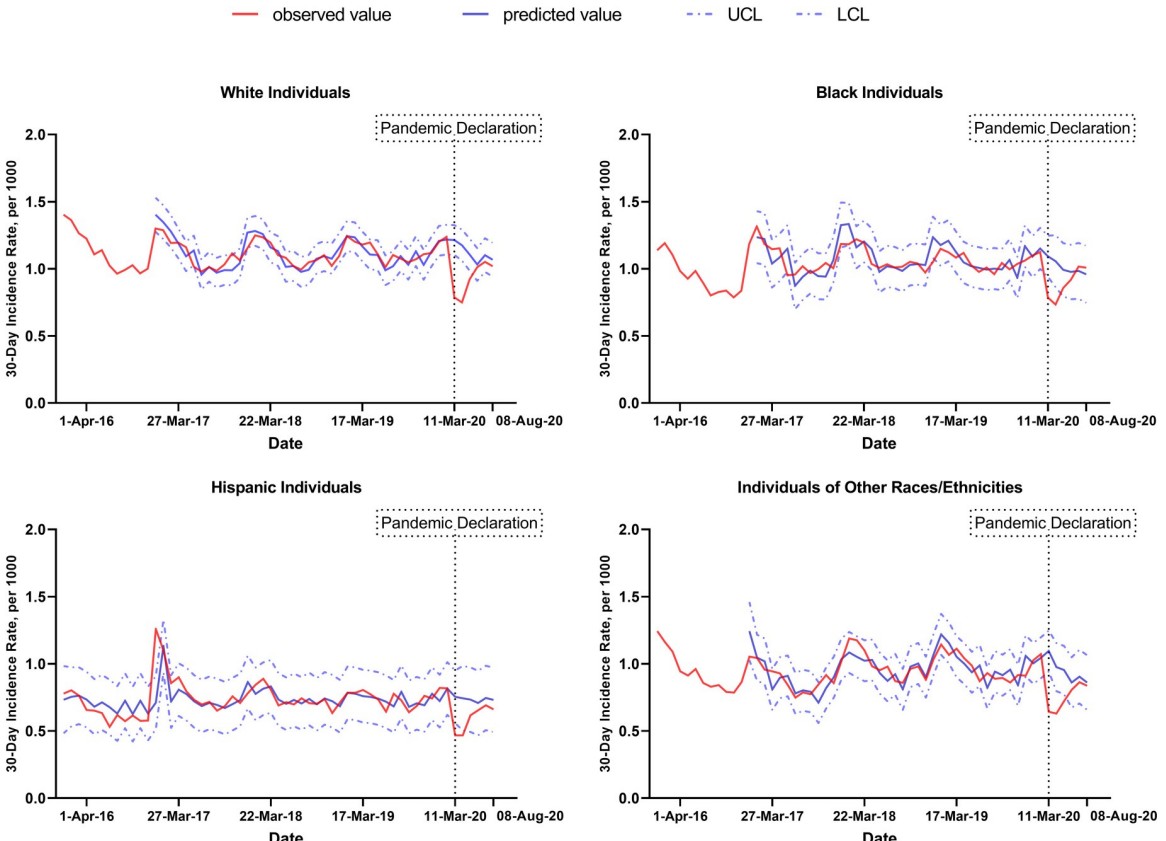

**Fig 3. Observed and predicted new atrial fibrillation diagnoses by race and ethnicity.** Abbreviations: UCL, upper confidence limit; LCL, lower confidence limit.

New AF diagnoses decreased across all states examined (**Fig 4**). There was no apparent relationship between the magnitude of the decrease of AF diagnoses and the cumulative number of COVID-19 cases in the first 30-day interval after pandemic declaration.

The left panel shows the cumulative number of COVID-19 cases per 100,000 state residents as of 04/09/2020 (with New York excluding New York City). The right panel shows the change in the rate of new atrial fibrillation diagnoses in the first 30-day interval after pandemic declaration (03/11/2020 to 04/09/2020), compared to the 30-day interval immediately before pandemic declaration (02/10/2020 to 03/10/2020). Only 28 states with a study sample size larger than 200,000 were examined, excluding 22 states and the District of Columbia.

## Discussion

In a large nationwide study, we observed that new AF diagnoses decreased immediately after the declaration of the COVID-19 pandemic; however, they returned to predicted levels by summer 2020. This decrease was consistent across racial and ethnic subgroups and states. The decrease in the rate of new AF diagnoses was larger for diagnoses originating from the outpatient setting than AF diagnoses originating from the inpatient setting.

A recent paper using Danish nationwide data observed a decrease in the number of new diagnoses of heart failure in the first four weeks after the COVID-19 pandemic declaration, when Denmark instituted a lockdown [26]. Even though the US did not impose a nationwide lockdown, our findings are consistent with those by Andersson et al. [26]. Previous US studies

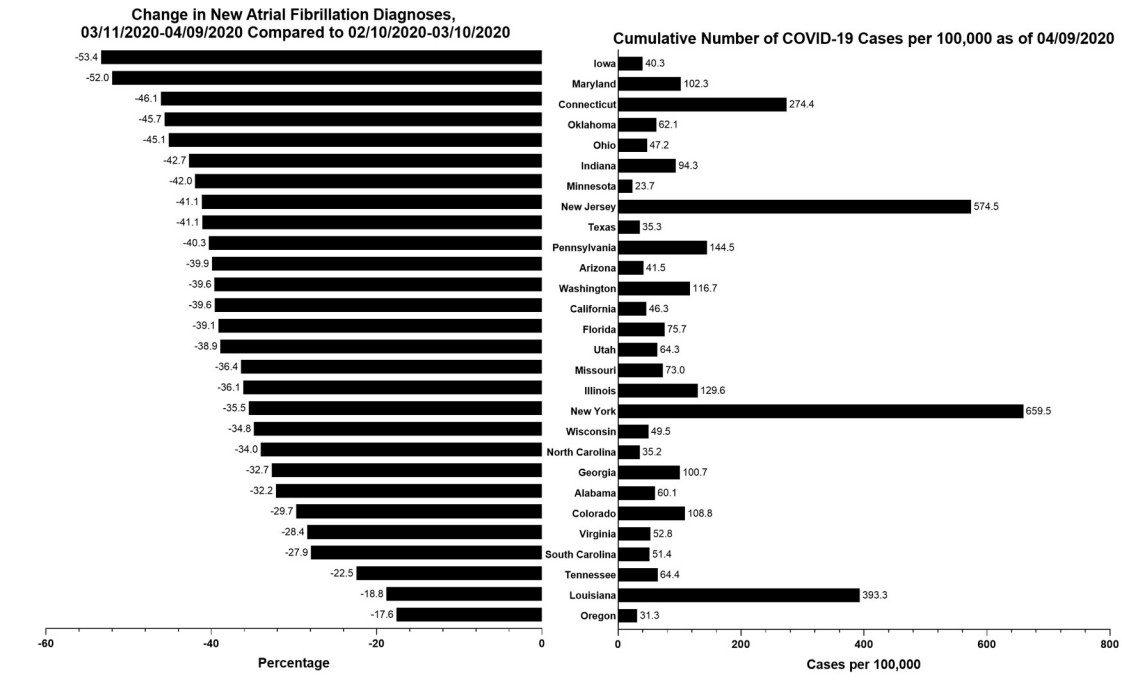

**Fig 4. Observed change in new atrial fibrillation diagnoses by state.**

have reported significant decreases in the number of hospitalizations and emergency room visits for common chronic conditions related to the pandemic, but have not evaluated new onset of chronic disease specifically, which was the objective of our study [2–5]. We observed that the magnitude of decrease in new AF diagnoses was lower in the inpatient setting and for ischemic stroke as initial manifestation of AF, which are more severe presentations of new-onset AF than those typically diagnosed in the outpatient setting. The less pronounced decrease in incidence of events observed for more severe outcomes is consistent with prior studies that reported that medical care avoidance in the early months of the COVID-19 pandemic was less pronounced for more severe events, including strokes [27, 28].

Notably, we observed no apparent relationship between the state-level magnitude of the decrease of AF diagnoses and the number of COVID-19 cases in the first 30-day interval after pandemic declaration. The lack of state variation may indicate that the decrease in AF diagnosis was more representative of patient's avoidance to seek medical care rather than the unavailability of providers and health care resources in hardest-hit areas. The behavioral changes in healthcare-seeking behavior may not have presented regional variation because of the nationwide reach of news reports on COVID cases in the early weeks of the COVID pandemic.

After the drop in new AF diagnosis observed immediately after the start of the COVID-19 pandemic, the incidence of AF nearly returned to predicted levels by summer 2020. We speculate that decreased diagnosis of AF may have represented fear of COVID-19 contagion in health care facilities, particularly in the first weeks after the declaration of the COVID-19 pandemic. It is possible that patients experiencing more frequent or severe symptoms decided to seek medical care after realizing that the risk of contagion would not cease in the short term, which would explain the return to expected levels within a few months of the pandemic onset. Nevertheless, these are speculative hypotheses as our data do not enable us to observe the reasons behind the changes in health care encounters noted in the study.

Interestingly, we observed a similar incidence of AF among White individuals and Black individuals. Racial differences in the incidence of AF have been observed in multiple studies, which have associated Black race with a lower incidence of AF [29–32]. This lower incidence, however, could represent ascertainment bias due to differential provider assessment of AF in Black individuals or lower access to care, which would limit the detection of AF [31, 33]. Under this hypothesis, it is likely that we observed similar rates of AF incidence across Black and White individuals because we used data from commercial insurance and Medicare Advantage beneficiaries, which represent a sample with higher income and access to medical care than the general population.

Our study has important implications for public health planning in future emergencies. We demonstrate that the effects of the COVID-19 pandemic extended beyond COVID-19 cases and deaths and evidences the nationwide impact of patients' avoidance of medical care in the earlier months of the COVID-19 pandemic. Delayed diagnosis of AF is concerning because pharmacotherapy with oral anticoagulation interestingand anti-arrhythmic therapy is available to prevent the increased risk of stroke associated with AF [34]; however, delayed diagnosis precludes timely treatment. As more data from the post-pandemic period become available, it will be relevant to assess the consequences of delayed diagnosis on clinical outcomes. For example, it will be important to evaluate whether the rate of ischemic stroke as initial manifestation of AF increases over time, as individuals that went undiagnosed with AF present with complications.

Our analysis is subject to some limitations. First, our data are limited to the first months after the onset of the COVID-19 pandemic. Although the lack of availability of more recent data is a limitation of our study, the time period captured is of major relevance for the study of pandemic disruptions of non-COVID disease due to lower access to care. This is because prior reports have demonstrated that patients avoidance of medical care was particularly profound in the early months of the COVID-19 pandemic [2–5]. Second, our analyses are based on claims data; therefore we were only able to capture medical events that triggered interactions with the health care system. Third, we did not conduct subgroup analyses for patients presenting with COVID-19, due to the inconsistent coding of COVID-19 cases in the first months of the pandemic. Finally, our sample is representative of commercial insurance and Medicare Advantage beneficiaries and therefore our findings are not generalizable to the overall population.

## Conclusions

In a nationwide cohort of 19.5 million individuals, new diagnoses of AF decreased substantially following the onset of the COVID-19 pandemic. Our findings evidence pandemic disruptions in access to care for AF, which are concerning because delayed diagnosis interferes with timely treatment to prevent serious complications.

## Supporting information

**S1 Table. Results of seasonal autoregressive integrated moving average models, primary analyses.** The estimated level change shows the immediate change in the outcome following the World Health Organization declaration of pandemic on 3/11/2020. The estimated trend change depicts the further change from the predicted every 30 days (slope).
(PDF)

**S2 Table. Results of seasonal autoregressive integrated moving average models, racial/ethnic subgroup analyses.** The estimated level change shows the immediate change in the

outcome following the World Health Organization declaration of pandemic on 3/11/2020. The estimated trend change shows the further change from the predicted every 30 days (slope).
(PDF)

**S3 Table. Results of interrupted time series analyses, racial/ethnic subgroup analyses.** The linear regression adjusted for autocorrelation using Newey-West standard error correction. Analysis was conducted using PROC MODEL with the SAS/ETS software. A joint Wald test shows no statistically significant difference in the change in the level of AF diagnoses after pandemic declaration across racial/ethnic subgroups (p = 0.34).
(PDF)

**S4 Table. Results of seasonal autoregressive integrated moving average models, sex subgroup analyses.** The estimated level change shows the immediate change in the outcome following the World Health Organization declaration of pandemic on 3/11/2020. The estimated trend change shows the further change from the predicted every 30 days (slope).
(PDF)

**S1 Fig. Selection of study sample.** The study cohort was selected using 2016—Q3 2020 data. Atrial fibrillation was defined as having a diagnosis claim with the ICD-9 code of 427.31 or ICD-10 codes of I48.0, I48.1, I48.2, or I48.91 in the first or second diagnosis field.
(PDF)

**S2 Fig. Observed and predicted new atrial fibrillation diagnoses by sex.** The upper panel shows trends in new atrial fibrillation diagnoses per 1000 female individuals for 30-day intervals, from 01/02/2016 to 09/06/2020. The lower panel shows trends in new atrial fibrillation diagnoses per 1000 male individuals for 30-day intervals, from 01/02/2016 to 09/06/2020. Solid red lines represent observed values. Solid blue lines represent predictions from ARIMA models the absence of the COVID-19 pandemic. In other words, they represent trends in AF diagnosis that would have been observed if there had not been a change in AF diagnoses following the COVID-19 pandemic. Dashed blue lines represent confidence intervals.
(PDF)

## Author Contributions

**Conceptualization:** Inmaculada Hernandez, Walid F. Gellad, Samir Saba, Emelia J. Benjamin, Jared W. Magnani.

**Data curation:** Gretchen Swabe.

**Formal analysis:** Meiqi He.

**Funding acquisition:** Inmaculada Hernandez, Walid F. Gellad, Utibe R. Essien, Samir Saba, Jared W. Magnani.

**Investigation:** Inmaculada Hernandez, Walid F. Gellad, Samir Saba, Emelia J. Benjamin, Jared W. Magnani.

**Methodology:** Inmaculada Hernandez, Meiqi He, Jingchuan Guo, Mina Tadrous, Nico Gabriel.

**Project administration:** Inmaculada Hernandez.

**Resources:** Inmaculada Hernandez.

**Supervision:** Inmaculada Hernandez.

**Writing – original draft:** Inmaculada Hernandez.

**Writing – review & editing:** Jingchuan Guo, Mina Tadrous, Nico Gabriel, Gretchen Swabe, Walid F. Gellad, Utibe R. Essien, Samir Saba, Emelia J. Benjamin, Jared W. Magnani.

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
