## [Decision Letter · Decision Letter 0]

9 Nov 2022

PONE-D-22-29038COVID-19 Pandemic and Trends in New Diagnosis of Atrial Fibrillation: A Nationwide Analysis of Claims DataPLOS ONE

Dear Dr. Hernandez,

Thank you for submitting your manuscript to PLOS ONE. After careful consideration, we feel that it has merit but does not fully meet PLOS ONE’s publication criteria as it currently stands. Therefore, we invite you to submit a revised version of the manuscript that addresses the points raised during the review process.

We look forward to receiving your revised manuscript.

Kind regards,

Han Eol Jeong, M.P.H., Ph.D.

Academic Editor

PLOS ONE

“I have read the journal's policy and the authors of this manuscript have the following competing interests: Hernandez has received consulting fees from Pfizer and Bristol Myers Squibb, outside of the submitted work.”

Reviewers' comments:

Reviewer's Responses to Questions

**Comments to the Author**

1. Is the manuscript technically sound, and do the data support the conclusions?

Reviewer #1: Yes

Reviewer #2: Yes

2. Has the statistical analysis been performed appropriately and rigorously? 

Reviewer #1: Yes

Reviewer #2: Yes

3. Have the authors made all data underlying the findings in their manuscript fully available?

Reviewer #1: Yes

Reviewer #2: Yes

4. Is the manuscript presented in an intelligible fashion and written in standard English?

Reviewer #1: Yes

Reviewer #2: Yes

5. Review Comments to the Author

Reviewer #1: The authors have conducted a population-based cohort study to quantify changes in new diagnoses of AF following the onset of the COVID-19 pandemic using a nationwide claims database. Although the research is interesting and well done, it needs further clarification in discussing and complementing the methodology.

Major comments

1. Introduction: I can understand that there is a problem in the diagnosis of chronic diseases due to the pandemic. However, the authors need additional logic as to why they studied AF.

2. Methods: Authors should present positive predictive value (PPV) together to enhance the reliability of AF diagnosis. In addition, to increase the validity of the definition of AF diagnosis, it is recommended to perform sensitivity analysis with AF event defined as Warfarin or NOAC prescription on the same date with AF diagnosis code.

3. Methods: Isn't it more appropriate to perform a stratified analysis considering the hospital setting, the number of hospitals by state, or the number of cardiologists, rather than analyzing by the state? Did the risk factors not affect AF diagnosis, except for ethnic differences?

4. Discussions: AF diagnosis decreases immediately after the pandemic declaration. However, it is soon observed to be similar to the predicted value. The author will need to offer a plausible explanation related to this part.

5. Discussions: Explain why the pattern of inpatient is different from the outpatient.

Minor comments

1. Methods; Outcomes: What is code 21 stand for? Authors should provide additional information on that code.

2. For the Hispanic population, a pick is observed in 2017, why is this observed?

3. Please add commas per 1000 units to Table 1

4. Please harmonize the date format between the manuscript and figure (e.g. 1/3/2019 or 1/Mar/2019)

5. Incorrect spacing appears throughout the manuscript. Again, please review in general.

Reviewer #2: For this study, the authors use data from a Optum’s Clinformatics® Data Mart database covering members of commercial insurance and Medicare beneficiaries to investigate the changes in incidence of new atrial fibrillation (AF) diagnosis following the COVID-19 pandemic. Outcome data were obtained from inpatient and outpatient claims record with primary and secondary diagnosis position. The authors compared the incidence rates of newly diagnosed AF per 30-day interval before and after the COVID-19 pandemic declaration (2020-03-11), and they tested changes and predicted trends in incidence rates of AF diagnosis using seasonal ARIMA model taking into account seasonality. Subgroup analyses by gender and race/ethnicity were performed and state variation in the changes in incidence rates of AF diagnoses were investigated. The authors noted substantial decrease in incidence rate of new AF diagnosis following the COVID-19 pandemic declaration, as supported by the additional analyses.

This study has addressed an important research question on the impact of pandemic on diagnosis of chronic disease. However, some of the data reported on changes in incidence rates of new AF diagnoses make question how adequately the magnitude of changes has been measured in this study. Providing relevant data would help strengthen this study.

My comments and suggestions for the authors are described in detail below.

Major Comment:

Comment 1. Assessment of changes in incidence rate of new AF diagnosis

a. Results for changes in 30-day incidence rate of new AF diagnosis (page 7, line 16-18 & 22-23; page 8, line 1 & 6-9; Figure 4; page 2, line 17-20). Considering the seasonality in AF diagnosis with higher incidence in winter, Jan to Feb (Figure 1-3), comparing the 30-day incidence rates immediately before and after COVID-19 pandemic declaration (2020-03-11) would like to make concerns the possibility of overestimation in measuring magnitude of change. The reviewer recommends comparing incidence rate of 30-day immediately after COVID-19 pandemic declaration with that of corresponding time-period in the previous year, or averaged in previous years, or predicted value, instead of immediately before 30-day incidence rate.

b. Based on the reported data (Figure 1 to 3), the incidence rate of AF diagnosis after COVID-19 pandemic declaration in summer (2020-08-08) does not seems to be largely different when compared with that of corresponding dates in previous years. Additional data on comparison of incidence rates in other time period after COVID-19 pandemic declaration with averaged incidence rates in corresponding time-period in previous years or predicted values (e.g., 31-60, 61-90, 91-120, 121-150, 151-180 days) would helpful to understand how long does the COVID-19 pandemic declaration or patients’ avoidance of medical care critically affect to the AF diagnosis, and strengthen the implication of this study.

Minor comment:

Comment 1. Subgroup analysis.

The authors note in the discussion that they provide evidence on the impact of patients’ avoidance of medical care in the earlier months of the COVID-19 pandemic (page 10). They further reported that Medicare Advantage beneficiaries were 84.4% of individuals newly diagnosed with AF, while 30.9% of study population were Medicare Advantage beneficiaries (Table 1). Higher proportion of Medicare Advantage beneficiaries in individuals newly diagnosed with AF may be because of that the incidence of AF is more common in elderly population, but it would be difficult to disregard differential patterns in the use of medical care by type of health insurance. Thus, it would be helpful to present data on the subgroup analysis by Medicare Advantage beneficiaries for readers to understand the effects of COVID-19 pandemic on the access to medical care for AF.

Comment 2. Discussion on state variation.

Please clarify what is meant by “There was no apparent relationship between the magnitude of the decrease of AF diagnoses and the cumulative number of COVID-19 cases in the first 30-day interval after pandemic declaration” (page 8). Does it mean that the changes in AF diagnoses immediately after pandemic declaration may have associated with the patients’ avoidance of medical care, rather than the accessibility? It would be helpful for reader to add to the discussion some consideration of why there was no apparent relationship between the magnitude of the changes in AF diagnoses and the number of COVID-19 cases.

Comment 3. Measurement of average 30-day incidence rate.

The authors note that “Before the onset of the COVID-19 pandemic, the estimated 30-day incidence rate of new AF diagnoses per 1000 individuals averaged 1.22 for White individuals, 1.15 for Black individuals, 0.82 for Hispanic individuals, and 1.05 for individuals of other races or ethnicities” (page 8). How long does the assessment period? From 2016-04-01 to 2020-03-11 OR 2016-01-01 to 2020-03-11? Considering the seasonality, it would be good to provide some detailed description in the Method section.

Comment 4. Structure of the Figure 4.

Suggest modifying the structure of the Figure 4 to improve readability as following: one y-axis of state in central, percentages of changes in new AF diagnoses in left-side, cumulative numbers of COVID-19 cases in right-side. Also, how about modify the order of states in y-axis in reverse order, from Oregon to Iowa, focusing on the magnitude of the changes in AF diagnoses?

6. PLOS authors have the option to publish the peer review history of their article (what does this mean?). If published, this will include your full peer review and any attached files.

Reviewer #1: No

Reviewer #2: No

---

## [Author Response · Author response to Decision Letter 0]

4 Jan 2023

We have ensured that our manuscript meets PLOS ONE's style requirements, including those for file naming.

“I have read the journal's policy and the authors of this manuscript have the following competing interests: Hernandez has received consulting fees from Pfizer and Bristol Myers Squibb, outside of the submitted work.”

Our conflicts do not alter our adherence to PLOS ONE policies on sharing data and materials.

We have included our updated statement of competing interests in the cover letter.

We obtained access to the data under a data user agreement that specifies that the sole proprietor of the raw data is Optum. This data user agreement explicitly prohibits the sharing of data to any individual who is not covered under the data user agreement. We are not able to make publicly available the minimal data set because of these legal restrictions.

We have included captions for Supporting Information files at the end of the manuscript (pages 18 and 19).

Reviewer #1:

The authors have conducted a population-based cohort study to quantify changes in new diagnoses of AF following the onset of the COVID-19 pandemic using a nationwide claims database. Although the research is interesting and well done, it needs further clarification in discussing and complementing the methodology.

Major comments

1. Introduction: I can understand that there is a problem in the diagnosis of chronic diseases due to the pandemic. However, the authors need additional logic as to why they studied AF.

We have added a sentence explaining why AF is a crucial disease state to measure COVID-19 disruptions of chronic disease care (page 3, lines 19-22): “AF is a critical disease state to measure the effects of the COVID-19 pandemic on non-COVID related chronic disease because every aspect of stroke prevention is vulnerable to disruption, including diagnosis, initiation of anticoagulation therapy, and treatment monitoring.”

2. Methods: Authors should present positive predictive value (PPV) together to enhance the reliability of AF diagnosis. In addition, to increase the validity of the definition of AF diagnosis, it is recommended to perform sensitivity analysis with AF event defined as Warfarin or NOAC prescription on the same date with AF diagnosis code.

Previous studies have estimated the positive predictive value of our definition to be 93-97%.1,2 We have incorporated these references in the methods section outcomes subsection (page 5, lines 1-2).

We did not perform sensitivity analyses requiring initiation of anticoagulation to define AF because claims data do not contain information on oral anticoagulation initiated during inpatient stays. This is because oral anticoagulants are not separately billed during inpatient stays. This would prevent us from identifying diagnosis in the inpatient setting according to the definition proposed by the reviewer. In addition, this definition does not consider the strong underutilization of oral anticoagulation in AF in the US, which has been well-described in the literature.3–5

3. Methods: Isn't it more appropriate to perform a stratified analysis considering the hospital setting, the number of hospitals by state, or the number of cardiologists, rather than analyzing by the state? Did the risk factors not affect AF diagnosis, except for ethnic differences?

Optum data do not enable us to identify the hospital where diagnosis took place, so we are not able to perform analyses stratifying by number of cardiologists in a hospital. We performed analysis stratified for inpatient and outpatient setting (page 9, lines 5-10, Figure 2). We do not follow the rationale for performing analyses based on the number of hospitals in each state. 

We did not incorporate other risk factors for AF in the analysis because the objective of our study was not to predict AF diagnosis but rather to understand changes in trends in AF diagnosis associated with the onset of the COVID-19 pandemic. Because the COVID-19 pandemic disproportionately affected underrepresented racial/ethnic groups, it was relevant to evaluate whether changes in diagnoses following pandemic declaration differed across racial/ethnic groups. We have incorporated this sentence in the introduction section (page 3 line 24, page 4 lines 1-2). 

4. Discussions: AF diagnosis decreases immediately after the pandemic declaration. However, it is soon observed to be similar to the predicted value. The author will need to offer a plausible explanation related to this part.

We believe that the immediate drop in diagnosis represents the fear of COVID-19 contagion in health care facilities, particularly in the first weeks after the declaration of the COVID-19 pandemic. It is possible that patients experiencing more frequent or severe symptoms decided to seek medical care after realizing that the risk of contagion would not cease in the short term. Nevertheless, this is a speculation as our data do not enable us to observe the reasons why AF diagnosis decreased. We have added a sentence in the first paragraph of the discussion section acknowledging that the incidence of new AF diagnoses returned to predicted levels in summer 2020 (page 11, lines 10-11): “In a large nationwide study, we observed that new AF diagnoses decreased immediately after the declaration of the COVID-19 pandemic; however, they returned to predicted levels by summer 2020”. We have incorporated the explanation provided above in the fourth paragraph of the discussion section, and we have acknowledged it is speculative and not based on findings from our study (page 12, lines 10-17): “After the drop in new AF diagnosis observed immediately after the start of the COVID-19 pandemic, the incidence of AF nearly returned to predicted levels by summer 2020. We speculate that decreased diagnosis of AF may have represented fear of COVID-19 contagion in health care facilities, particularly in the first weeks after the declaration of the COVID-19 pandemic. It is possible that patients experiencing more frequent or severe symptoms decided to seek medical care after realizing that the risk of contagion would not cease in the short term, which would explain the return to expected levels within a few months of the pandemic onset. Nevertheless, these are speculative hypotheses as our data do not enable us to observe the reasons behind the changes in health care encounters noted in the study”. 

5. Discussions: Explain why the pattern of inpatient is different from the outpatient.

We discuss in the second paragraph of the discussion section why the decrease in new AF diagnoses was less pronounced in the inpatient setting (page 11 lines 21-24, page 12 lines 1-2): “We observed that the magnitude of decrease in new AF diagnoses was lower in the inpatient setting and for ischemic stroke as initial manifestation of AF, which are more severe presentations of new-onset AF than those typically diagnosed in the outpatient setting. The less pronounced decrease in incidence of events observed for more severe outcomes is consistent with prior studies that reported that medical care avoidance in the early months of the COVID-19 pandemic was less pronounced for more severe events, including strokes”.

Minor comments

1. Methods; Outcomes: What is code 21 stand for? Authors should provide additional information on that code. 

Thank you for seeking clarification. Place of service 21 represents inpatient hospital, which we now state in the methods (page 5, lines 11-12).

2. For the Hispanic population, a pick is observed in 2017, why is this observed?

The peak coincides with the beginning of the year, which is when the majority of beneficiaries are added to the sample, as the enrollment in health plans often starts on January 1. Similar peaks are observed every year in January for the overall population. We hypothesize that the 2017 peak is particularly pronounced for the Hispanic subgroup because the lower sample size of this ethnic group makes the trend particularly sensitive to enrollment changes. 

3. Please add commas per 1000 units to Table 1

Commas have been added.

4. Please harmonize the date format between the manuscript and figure (e.g. 1/3/2019 or 1/Mar/2019)

We have not made this change as we are unsure of the date format preferred by PLOS ONE. We believe this change can be handled at the proofing / editing stage. 

5. Incorrect spacing appears throughout the manuscript. Again, please review in general.

We have revised the manuscript to ensure correct spacing.

Reviewer #2

For this study, the authors use data from a Optum’s Clinformatics® Data Mart database covering members of commercial insurance and Medicare beneficiaries to investigate the changes in incidence of new atrial fibrillation (AF) diagnosis following the COVID-19 pandemic. Outcome data were obtained from inpatient and outpatient claims record with primary and secondary diagnosis position. The authors compared the incidence rates of newly diagnosed AF per 30-day interval before and after the COVID-19 pandemic declaration (2020-03-11), and they tested changes and predicted trends in incidence rates of AF diagnosis using seasonal ARIMA model taking into account seasonality. Subgroup analyses by gender and race/ethnicity were performed and state variation in the changes in incidence rates of AF diagnoses were investigated. The authors noted substantial decrease in incidence rate of new AF diagnosis following the COVID-19 pandemic declaration, as supported by the additional analyses.

This study has addressed an important research question on the impact of pandemic on diagnosis of chronic disease. However, some of the data reported on changes in incidence rates of new AF diagnoses make question how adequately the magnitude of changes has been measured in this study. Providing relevant data would help strengthen this study.

My comments and suggestions for the authors are described in detail below.

Major Comment:

Comment 1. Assessment of changes in incidence rate of new AF diagnosis

a. Results for changes in 30-day incidence rate of new AF diagnosis (page 7, line 16-18 & 22-23; page 8, line 1 & 6-9; Figure 4; page 2, line 17-20). Considering the seasonality in AF diagnosis with higher incidence in winter, Jan to Feb (Figure 1-3), comparing the 30-day incidence rates immediately before and after COVID-19 pandemic declaration (2020-03-11) would like to make concerns the possibility of overestimation in measuring magnitude of change. The reviewer recommends comparing incidence rate of 30-day immediately after COVID-19 pandemic declaration with that of corresponding time-period in the previous year, or averaged in previous years, or predicted value, instead of immediately before 30-day incidence rate.

Thank you for the suggestion. The incidence in new AF diagnosis in the first 30-day period after pandemic declaration (3/11/2020-4/9/2020) was 0.74, this is 35% lower than in the period immediately before pandemic declaration (incidence rate=1.14). The average incidence rate in the March periods from previous years was 1.134. Using this estimate as reference, the estimated decrease in AF diagnosis is 34.7%, which rounds to 35%. We have not made the change in the manuscript as our method does not lead to an overestimation of the magnitude of change.

b. Based on the reported data (Figure 1 to 3), the incidence rate of AF diagnosis after COVID-19 pandemic declaration in summer (2020-08-08) does not seems to be largely different when compared with that of corresponding dates in previous years. Additional data on comparison of incidence rates in other time period after COVID-19 pandemic declaration with averaged incidence rates in corresponding time-period in previous years or predicted values (e.g., 31-60, 61-90, 91-120, 121-150, 151-180 days) would helpful to understand how long does the COVID-19 pandemic declaration or patients’ avoidance of medical care critically affect to the AF diagnosis, and strengthen the implication of this study.

We have modified the results section following this suggestion (page 8, lines 9-12): “The observed incidence of new AF diagnosis was substantially lower than predicted by regression models in the first 90 days after the declaration of the COVID-19 pandemic, however these differences closed by summer 2020. As of June 2020, the observed incidence of new AF diagnoses was only 6% lower than predicted.” 

We have added a sentence in the first paragraph of the discussion section acknowledging that the incidence of new AF diagnoses returned to predicted levels in summer 2020 (page 11, lines 10-11): “In a large nationwide study, we observed that new AF diagnoses decreased immediately after the declaration of the COVID-19 pandemic; however, they returned to predicted levels by summer 2020”. We have also added a paragraph in the discussion section commenting on this important finding, as suggested by both reviewers (page 12, lines 10-17): “After the drop in new AF diagnosis observed immediately after the start of the COVID-19 pandemic, the incidence of AF nearly returned to predicted levels by summer 2020. We speculate that decreased diagnosis of AF may have represented fear of COVID-19 contagion in health care facilities, particularly in the first weeks after the declaration of the COVID-19 pandemic. It is possible that patients experiencing more frequent or severe symptoms decided to seek medical care after realizing that the risk of contagion would not cease in the short term, which would explain the return to expected levels within a few months of the pandemic onset. Nevertheless, these are speculative hypotheses as our data do not enable us to observe the reasons behind the changes in health care encounters noted in the study”. 

Minor comments:

Comment 1. Subgroup analysis.

The authors note in the discussion that they provide evidence on the impact of patients’ avoidance of medical care in the earlier months of the COVID-19 pandemic (page 10). They further reported that Medicare Advantage beneficiaries were 84.4% of individuals newly diagnosed with AF, while 30.9% of study population were Medicare Advantage beneficiaries (Table 1). Higher proportion of Medicare Advantage beneficiaries in individuals newly diagnosed with AF may be because of that the incidence of AF is more common in elderly population, but it would be difficult to disregard differential patterns in the use of medical care by type of health insurance. Thus, it would be helpful to present data on the subgroup analysis by Medicare Advantage beneficiaries for readers to understand the effects of COVID-19 pandemic on the access to medical care for AF.

Thank you for the comment. Optum data includes information on beneficiaries of commercial plans and of Medicare Advantage plans offered by a single insurer (UnitedHealthcare). All Medicare beneficiaries included in the analyses are enrolled in Medicare Advantage (no Medicare fee-for-service beneficiaries). For this reason, it is not possible to differentiate between the effect of age and type of health plan. The primary findings of the study are driven by Medicare beneficiaries (as opposed to commercial beneficiaries) because atrial fibrillation is an age-related disease. We have not added a subgroup analysis for Medicare beneficiaries because: 1) the subgroup analyses do not allow a differentiation between effects of age eligibility for Medicare enrollment and true differences in insurance design; 2) the differences in insurance factors as they relate to AF diagnosis are minimal as both insurance products are offered by the same carrier.

Comment 2. Discussion on state variation.

Please clarify what is meant by “There was no apparent relationship between the magnitude of the decrease of AF diagnoses and the cumulative number of COVID-19 cases in the first 30-day interval after pandemic declaration” (page 8). Does it mean that the changes in AF diagnoses immediately after pandemic declaration may have associated with the patients’ avoidance of medical care, rather than the accessibility? It would be helpful for reader to add to the discussion some consideration of why there was no apparent relationship between the magnitude of the changes in AF diagnoses and the number of COVID-19 cases.

In performing these analyses, we hypothesized that decreases in new AF diagnosis would be larger in states with a higher burden of COVID cases because of patients’ avoidance of medical care and because of lower availability of providers and health care resources for non-COVID-disease. However, this hypothesis was not confirmed by our analyses. We have added a new paragraph in the discussion section commenting this finding, as requested by the reviewer (page 12, lines 3-9): “Notably, we observed no apparent relationship between the state-level magnitude of the decrease of AF diagnoses and the number of COVID-19 cases in the first 30-day interval after pandemic declaration. The lack of state variation may indicate that the decrease in AF diagnosis was more representative of patient’s avoidance to seek medical care rather than the unavailability of providers and health care resources in hardest-hit areas. The behavioral changes in healthcare-seeking behavior may not have presented regional variation because of the nationwide reach of news reports on COVID cases in the early weeks of the COVID pandemic.”

Comment 3. Measurement of average 30-day incidence rate.

The authors note that “Before the onset of the COVID-19 pandemic, the estimated 30-day incidence rate of new AF diagnoses per 1000 individuals averaged 1.22 for White individuals, 1.15 for Black individuals, 0.82 for Hispanic individuals, and 1.05 for individuals of other races or ethnicities” (page 8). How long does the assessment period? From 2016-04-01 to 2020-03-11 OR 2016-01-01 to 2020-03-11? Considering the seasonality, it would be good to provide some detailed description in the Method section.

We apologize for the confusion; these statistics refer to the period immediately before the declaration of the COVID-19 pandemic (2/10/2020-3/10/2020). We have now added this information to the text (page 10, line 1).

Comment 4. Structure of the Figure 4.

Suggest modifying the structure of the Figure 4 to improve readability as following: one y-axis of state in central, percentages of changes in new AF diagnoses in left-side, cumulative numbers of COVID-19 cases in right-side. Also, how about modify the order of states in y-axis in reverse order, from Oregon to Iowa, focusing on the magnitude of the changes in AF diagnoses?

We appreciate the suggestion and have reformatted the figure as requested.

References

1. Rix TA, Riahi S, Overvad K, Lundbye-Christensen S, Schmidt EB, Joensen AM. Validity of the diagnoses atrial fibrillation and atrial flutter in a Danish patient registry. Scand Cardiovasc J. 2012;46(3):149-153.

2. Sundbøll J, Adelborg K, Munch T, et al. Positive predictive value of cardiovascular diagnoses in the Danish National Patient Registry: a validation study. BMJ Open. 2016;6(11):e012832.

3. Hernandez I, He M, Chen N, Brooks MM, Saba S, Gellad WF. Trajectories of oral anticoagulation adherence among Medicare beneficiaries newly diagnosed with atrial fibrillation. J Am Heart Assoc. 2019;8(12):e011427.

4. Dhamane AD, Hernandez I, Di Fusco M, et al. Non-persistence to oral anticoagulation treatment in patients with non-valvular atrial fibrillation in the USA. Am J Cardiovasc Drugs. 2022;22(3):333-343.

5. Hernandez I, Saba S, Zhang Y. Geographic variation in the use of oral anticoagulation therapy in stroke prevention in atrial fibrillation. Stroke. 2017;48(8):2289-2291.

---

## [Editor Report · Decision Letter 1]

16 Jan 2023

COVID-19 Pandemic and Trends in New Diagnosis of Atrial Fibrillation: A Nationwide Analysis of Claims Data

PONE-D-22-29038R1

Dear Dr. Hernandez,

We’re pleased to inform you that your manuscript has been judged scientifically suitable for publication and will be formally accepted for publication once it meets all outstanding technical requirements.

Kind regards,

Han Eol Jeong, M.P.H., Ph.D.

Academic Editor

PLOS ONE
---

## [Editor Report · Acceptance letter]

20 Jan 2023

PONE-D-22-29038R1 

COVID-19 pandemic and trends in new diagnosis of atrial fibrillation: A nationwide analysis of claims data 

Dear Dr. Hernandez:

I'm pleased to inform you that your manuscript has been deemed suitable for publication in PLOS ONE. Congratulations! Your manuscript is now with our production department. 

Kind regards, 

on behalf of

Dr. Han Eol Jeong 

Academic Editor

PLOS ONE